# Nutrients and Caloric Intake Associated with Fruits, Vegetables, and Legumes in the Elderly European Population

**DOI:** 10.3390/nu12092746

**Published:** 2020-09-09

**Authors:** Danila Azzolina, Luca Vedovelli, Silvia Gallipoli, Megan French, Marco Ghidina, Manfred Lamprecht, Melina Tsiountsioura, Giulia Lorenzoni, Dario Gregori

**Affiliations:** 1Unit of Biostatistics, Epidemiology, and Public Health, Department of Cardiac, Thoracic, and Vascular Sciences, University of Padova, 35143 Padova, Italy; danila.azzolina@uniupo.it (D.A.); luca.vedovelli@unipd.it (L.V.); giulia.lorenzoni@unipd.it (G.L.); 2Research Support Unit, Department of Translational Medicine, University of Eastern Piedmont, 28100 Novara, Italy; 3Zeta Research, 34122 Trieste, Italy; silviagallipoli@zetaresearch.com (S.G.); meganfrench@zetaresearch.com (M.F.); MarcoGhidina@zetaresearch.com (M.G.); 4The Juice Plus+® Science Institute, Collierville, TN 38017, USA; Manfred.Lamprecht@Juiceplus.com (M.L.); melina.tsiountsioura@greenbeat.at (M.T.); 5Otto Loewi Research Center, Division of Physiological Chemistry, Medical University of Graz, 8010 Graz, Austria; 6Green Beat—Institute of Nutrient Research, 8010 Graz, Austria

**Keywords:** nutrition, fruit, vegetables, non-communicable disease, elders, policies

## Abstract

Nutrition is emerging as a key factor in promoting healthy lifestyles in the growing elderly population across Europe. In this study, we examined the non-animal-derived food source consumption among the elderly European population to evaluate the actual contributions of these foods to the diet of the elders. We gathered 21 studies conducted in 17 European countries to evaluate the fruit, vegetable, and legume (along with their derived products) consumption among the elderly (>65 years) population. Foods’ nutritional values were calculated and compared to the recommended intakes. A Bayesian multilevel hierarchical analysis was conducted to estimate the caloric intake of food categories and to compare the elderly and general adult populations. Although the lowest consumption was generally associated with the lowest nutrient and fiber intake, the reverse was not always the case. Concerning the general adult population, no differences in the related caloric intake of elders were noticed. Differences were instead evident when foods were divided into subclasses. Elderly populations consume fruit and fruit products, but they drink less fruit and vegetable juices and nectars. In conclusion, elderlies’ fruit and vegetable consumption showed a peculiar pattern with respect to the general adult population, whose recognition could be helpful to address tailored policies. Constantly updated studies, including all the lifespan ages, are warranted to design tailored effective public health interventions.

## 1. Introduction

The World Health Organization (WHO) Global Status Report on non-communicable diseases (NCDs) 2010 [1] reported that 63% of the 57 million global deaths in 2008 were due to NCDs, with the four major NCDs being cardiovascular disease (CVD), diabetes, cancer, and chronic respiratory diseases (CRD). NCDs are preventable through reductions of the four main behavioral risk factors: tobacco use, physical inactivity, harmful use of alcohol, and unhealthy diet [1]. In 2017, 11 million deaths and 255 million disability-adjusted life years (DALYs) were attributable to dietary risk factors, with diets low in fruits being the third-highest dietary risk factor, accounting for two million deaths and 65 million DALYs, and diets low in vegetables being the fifth-highest dietary risk factor [2].

In 2003, the WHO and Food and Agriculture Organization (FAO) [3] reported convincing associations between fruit and vegetable (F&V) consumption and reduced risk of CVD and a protective association between coronary heart disease (CHD) and stroke. F&V is one of the most diverse food groups providing nutrients [4] and is generally considered a good source of micronutrients, minerals, fiber, and phytochemicals [5,6]. However, the beneficial health effects are not generally explained by individual nutritional components but rather by their combinations along with their low energy density and high fiber [4]. A diet containing more potassium, calcium, and magnesium sourced from F&V, low or non-fat dairy products, and less overall fat, has, for example, been shown to lower blood pressure [6]. Other studies have also shown that increased potassium intake may help decrease blood pressure [7,8] and is associated with a decreased risk of stroke [8,9] and possibly other CVDs [10,11,12,13,14]. The risk of CHD and stroke have also been reported as being lowered by increased dietary fiber [11,15].

Nutrition from F&V also plays a role in preventing anemia through the intake of iron and folic acid, which are found in some F&V and legumes [12]. Folic acid also has a preventative effect on neural tube defects and may contribute to reducing the risk of CVD [12]. Nutrition from F&V may also influence reducing the risk of dementia [13] and supports the maintenance of bone mineral density [14]. High potassium intake, for example, has been demonstrated to have protective effects on bone maintenance and several pathologic states, including the cardiovascular system and kidneys [15]. Moderate evidence indicates that healthy dietary patterns—including those higher in F&V—reduce the risk of developing type 2 diabetes mellitus [16]. Increased consumption of F&V, along with increased physical activity, are considered key strategies for stemming the epidemic of obesity and associated diseases, partly concerning fiber intake [3]. The fiber content of F&V is understood to play a role in controlling cholesterol and blood sugar levels (soluble fiber) and in preventing constipation (insoluble fiber) and is also understood to have a preventative effect against colon and breast cancer [12]. The WHO states that F&V intake is a probable protective factor for cancer, with overweight and obesity appearing to be the most important known avoidable causes of cancer after tobacco use [3]. The International Agency for Research on Cancer (IARC) discusses at length the strength of the evidence of F&V and cancer preventative effects [17]. Particular interest has been given to the preventative role of non-nutrient phytochemicals (e.g., phenolics, such as flavonoids), which are believed to be effective in a synergistic manner [18] and have multiple mechanisms of action that extend beyond antioxidant activity. Some phytochemicals have been shown to have preventative effects against cancer and CVD [19], and recommendations have been made for increased consumption of antioxidant-rich F&V for the prevention of diseases [20].

Elderly populations are considered more vulnerable to inadequate nutrition than younger adults [21], with nutritional intake in the elderly population being a function of medical, social, environmental, functional, and economic factors [21,22]. Increased demand for health services is associated with poor nutritional status, which is also recognized as an important predictor of morbidity and mortality [21]. An estimated 5–10% of independently living elderly individuals, and 30–60% of those hospitalized or institutionalized [23], are understood to be undernourished, which increases the risk of numerous medical conditions [21]. Greater consumption of F&V by institutionalized elderly populations has been associated with increased vitamin and mineral intake [24]. In addition to the previously mentioned points for the general population, the importance of F&V consumption for elderly people is illustrated by findings that show how F&V consumption—along with supplemental intake of potassium and magnesium—has been found to contribute to the maintenance of bone mineral density in elderly subjects [14].

In Europe, the F&V products are, generally, well appreciated and consumed by the elderly subjects [25]. However, the consumption is still insufficient for the European elderly people, and in several cases, below the WHO recommended 400g per day [5,12]. Moreover, a certain heterogeneity in the F&V intake among the European countries has been evidenced; the F&V consumption is generally at higher levels in the Southern European countries compared with the other regions [26].

Despite these results, little effort has been made in the literature to better characterize the F&V consumption among the elderly people as compared to the adult population.

In this study, we aimed to determine the nutrients consumed from fruits, vegetables, and legumes by European self-sufficient elderly populations (65–74 years), and to compare the contributions of total nutrients from these foods to recommended daily intakes for all foods. We further aimed to determine the nutrients received from certain fruits, vegetables, and legumes in average elderly individual diets.

## 2. Materials and Methods

### 2.1. Data Sources

Food consumption can be estimated through food consumption surveys (i.e., records/diaries, food frequency questionnaires, dietary recall, and total diet studies) at an individual or household level or can be approximated through food production statistics [27], for example, based on an FAO Food Balanced Sheet (FBS) that represents foods available for consumption by the whole population. General information regarding FAO FBS (including methods, criticisms, and potential sources of error) household budget surveys and individual dietary surveys are discussed and detailed elsewhere [4].

The main source of availability data is within the WHO Global Environment Monitoring System/Food Contamination Monitoring and Assessment Programme (GEMS/Food) [28], which is based on the FAO 2002–2007 FBS [29] (i.e., quantities produced and imported minus quantities exported and used as animal feed or for seed) from 179 countries clustered into 17 diets (G01–G17) based on statistical similarities between dietary patterns (of 20 key foods, all food groups). The 2012 GEMS database [30] contains availability (grams/capita/day, g/capita/d) data for the main food groups (citrus fruit, pome fruits, etc.) for G01–G17. This database was used to develop the International Estimated Daily Intake (IEDI) Food consumption cluster diets database and tool (version 3, 2018) [31], which includes consumption data (g/d) at a more detailed individual commodity level. After the main group (e.g., pome fruits), sub-totals are made in decreasing levels depending on the commodity; for example, level 1 pome fruits; level 2 “group of pome fruits, raw (incl. apple juice, incl. apple cider)”; level 3 “apple, raw (incl. juice, incl. cider)”, “loquat, raw (incl. processed)”, “pear, raw”; etc. The database tool was developed to compute daily intakes as chronic exposure to contaminants from diet consumption data for the WHO food safety chemical risks.

Specific F&V consumption data for Europe are available through the European Food Safety Authority (EFSA) database, which was built from existing national information on food consumption at a detailed level. Organizations within the European Union have provided EFSA with data from their national dietary surveys, resulting in a compilation of 53 different dietary surveys from 22 different EU Member States [32]. Data (median, percentiles, mean, and standard deviation) are presented by age group according to the food categorization tier system of seven exposure hierarchies (e.g., level 1 “vegetables and vegetable products”; level 2 “leafy vegetables”; level 3 “leafy brassica”; level 4 “Chinese cabbages and similar”; level 5/6/7 “Chinese cabbage”) of the EFSA standardized food classification and description system (FoodEx2) [28] for both acute and chronic consumption (consumer or all subjects) and are shown as either g/d or g/kg body weight per day. Only data obtained through food records, 24-h dietary recall, and 48-h dietary recall are included in the EFSA database. Further details of the database and methodologies are provided in EFSA publications [33].

### 2.2. Data Extraction and Calculations

The IEDI food consumption cluster diet database [31] rearranged WHO GEMS 17 cluster diet food availability data (FAO FBS) [30] according to codex codes and recalculated the data to individual commodities. As mentioned previously, 179 countries are allocated to one of the geographical clusters G01–G17 (Table 1). Most European countries fall into G06, G07, G08, G10, G11, or G15. Availability data (mean g/capita/d) were extracted from the IEDI Food consumption cluster diet database [31] for fruit, vegetable, legume, and starchy root/tuber products at either subgroup level 2 (e.g., “citrus fruit”/”subgroup of mandarins, raw (incl. mandarin juice)”) or level 3 (e.g., “berries and other small fruits”/”cane berries”/”blackberries, raw”) depending on the product. In cases where products included those for alcohol (i.e., grapes and apples), oil (e.g., olives and soya bean), sugar (i.e., sugar beet), or flour (e.g., potatoes and cassava) production, the product was selected to exclude these elements (e.g., “apple, raw (incl. juice, excl. cider)” instead of “apple, raw (incl. juice, incl. cider)”).

We assigned each of the 159 extracted individual products to relevant hierarchy levels 1–3 used by EFSA (e.g., “apple, raw (incl. juice, excl. cider)” assigned to EFSA level 3 “pome fruit”; level 2 “fruit used as fruit”; level 1 “fruit and fruit products”), such that available data were sorted to agree with consumption data, with the exception of fruit and vegetable juice, which was extracted from IEDI along with the main product (e.g., apple juice with apples) since not all juices are defined in IEDI (e.g., carrot juice). The total availability (g/capita/d) of fruit (including juice), total availability of vegetables (including juice), and total availability of legumes were calculated by adding the individual products in these groups for G01–G17; finally, total availability of fruits, vegetables, and legumes was calculated, with the option of adding starchy roots/tubers.

Individual products (raw) in the IEDI/GEMS database were matched with the nearest similar (raw/uncooked where available) product in the Public Health England McCance and Widdowson composition of foods database [34] or, when not available, the US food composition online database [35]. Estimated energy (kcal), dietary fiber (AOAC), and 19 select micronutrients (potassium, calcium, magnesium, phosphorus, iron, copper, zinc, manganese, selenium, vitamin A/C/E/K/B1 thiamine/B2 riboflavin/B3 niacin/B6/B5 pantothenic acid/B9 folate) were calculated from the grams/day of each product multiplied by each nutrient unit measurement per 100 g divided by 100 to yield each nutrient unit measurement/capita/day (e.g., banana potassium, K, the content of 208 mg K/100 g, and availability G01 of 5.25 g/person/d, yields 10.9 mg K/person/d). If vitamin A as retinol activity equivalents (RAE) was not defined for a given commodity, but beta-carotene or retinol equivalents (RE) were, RAE was assumed first as beta-carotene divided by 12. If beta-carotene data were not given, RAE was assumed as RE/2. In some instances, neither were available, and RAE was assumed as zero [36]. Nutrients were sub-totaled at hierarchy level 1 for each of the geographic clusters (G01-G17), and sub-totals at level 1 were added as total mean estimated nutrient availability from fruits, vegetables, and legumes. These totals for each geographic cluster were divided by the US recommended daily allowance (RDA) or adequate intake (AI, Table 2) for adult males and multiplied by 100 to express the percentage contribution of estimated available fruits, vegetables, and legumes to the RDA for fiber and each of the 19 nutrients.

### 2.3. Consumption Data and Calculations

EFSA chronic consumption data (g/person/d, all subjects) from 21 (65–74 years) studies in elderly populations across 17 countries (Appendix A) were used in this study. The elderly mean consumption of fruit, vegetables, legumes across the considered study has been also reported in the Appendix A. Data were downloaded at hierarchy level 7 (e.g., level 7/6/5 apples; level 4 apples and similar; level 3 pome fruits; level 2 fruits used as fruit; level 1 fruit and fruit products) for the following level 1 E FSA groups: fruit and fruit products; vegetables and vegetable products (includes legumes with pods); fruit and vegetable juices and nectars (including concentrates); legumes, nuts; oilseeds and spices. Nuts, oilseeds, and spices were excluded, except dried herbs, because fresh herbs are included within the vegetable category. Thus, the legumes category here actually refers to legumes without pods and dried herbs. Each product at level 7 remained associated with the corresponding hierarchy levels 1–6. The mean consumption (g/person/d) for each elderly population under study was totaled using data at level 7 for each level 1 category: fruit, vegetable, fruit and vegetable juice, and legumes.

Each EFSA product at level 7 was matched with the nearest similar product in the Public Health England McCance and Widdowson composition of foods database [34] or, when not available, the US food composition online database [35]. Nutrient data for cooked products (generally as boiled where available) were selected for products that are mostly eaten cooked, e.g., eggplant; otherwise, nutrient data for the raw or processed product were used, e.g., lettuce. For processed or lesser common products that were not listed in the Public Health England nutrient database and thus taken from the US nutrient conversion database, only basic data were generally available. For the parameters determined here, data were often only available for fiber, vitamin C, and RAE. Therefore, although other nutrients are absent in the calculations, they have a contribution (e.g., pureed fruits and vegetables, sun-dried tomatoes, and sweet peppers). We avoided this potential problem for processed juice products (mango nectar, apricot nectar, etc.) that are not listed in the Public Health England nutrient database, and for which limited data are available in the US database, by assuming them as their unprocessed equivalent (e.g., apricot fruits instead of apricot nectar) to obtain an entire set of nutrients. Estimated energy (kcal), dietary fiber, and 19 select nutrients were determined as described before. Nutrients were sub-totaled at hierarchy level 1 for each of the 21 studies and were then added as total mean estimated nutrient consumption from fruits and vegetables (including juice and legumes) for each study. These totals for each study were divided by the US RDA or AI with male adults for adult population data and males aged 70 and over for elderly population data.

### 2.4. Caloric Intake Estimate from Fruits and Vegetables

A Bayesian multilevel hierarchical model was carried out to estimate the caloric intake for fruit and vegetable products across European countries, stratifying the model estimates according to age classes (adult versus elderly). The basic version of a Bayesian model follows the form:*yi∼Normal(θi,σi)*(1)
where *yi* is the point estimate for the caloric intake related to the single country *i*, which is presumed to have been a draw from a normal distribution centered on *θi*. The standard error for the specific country estimate is *σi*, which is the standard deviation of the normal distribution. The model assumes a random effect term on the country level caloric intake distributed as:*θi∼Normal(μ,τ)*(2)
where μ is the mean caloric intake without considering country-level effect influenced by age classes (adult vs. elderly), and τ is the variability (heterogeneity) around that mean.
*μ = α + βElderly*(3)

Uninformative priors have been considered for the analysis:*μ∼Uniform(−∞,∞)*(4)
*τ∼Uniform(0,1000)*(5)

The data presented in this paper have been made available to the public on the website www.round-project.com (see Appendix A for details). The computations were performed in 4 chains and 2000 iterations using the brms [37] package in R [38] interfacing with Stan. The model convergence was assessed by performing a visual inspection of the trace plot diagram. R Software 4.0 was used for the analysis.

## 3. Results

### 3.1. IEDI GEMs Elderly Availability

The IEDI GEMs availability data [31] for geographical clusters corresponding to European countries with total fruits, vegetables, and legumes are shown in Table 3 and Table 4. The lowest mean availability was 584.3 g/person/d for G07 (e.g., Finland, France, and the United Kingdom), and the highest was 652.8 g/person/d for G11 (Belgium and The Netherlands).

### 3.2. IEDI GEMs Elderly Nutrition Conversion

Estimated daily availability of fiber and nutrients from total fruits, vegetables, and legumes to RDAs or AIs for European countries are shown in Table 3. Only vitamin C and partially vitamin K1 passed the recommended RDA for males older than 70 years.

Means of European G cluster availability of fruits, vegetables, legumes, and associated nutrients (as a percentage contribution to US RDA or AI for males >70 years) are available in Appendix A.

### 3.3. EFSA Elderly Consumption

The data for elderly (65–74 years) populations are shown in Figure 1 and illustrate that 43% of studies (9/21) involving elderly populations resulted in mean consumption data below the WHO recommended daily intake of 400 g/d, with the average of all the EFSA database elderly surveys being 415.4 g/d, ranging from 287.2 g/d (Austria 2010, N = 67) to 575.3 g/d (France 2014, N = 384).

### 3.4. EFSA Elderly Nutrition Conversion

Estimated daily consumption of fiber and nutrients from total fruits, vegetables, and legumes for EFSA elderly population data with the estimated percentage contributions of these data to AIs and RDAs are shown in Table 4 (see Appendix A for the non-pooled data of males and females). The average energy from fruit, vegetables, and legumes consumed for EFSA studies in elderly populations was 199.7 kcal/person/d (range 142.3 kcal/d for Portugal in 2015, to 390.0 kcal/d for Germany in 2007). Only vitamin C (partially) passed the recommended RDA for males older than 70 years.

Austria in 2011 reported low percentage contributions of fiber, potassium, calcium, magnesium, phosphorus, iron, copper, zinc, RAE, vitamin E, thiamine, riboflavin, niacin, vitamin B6, folate, and pantothenate, while also having the lowest consumption of the same nutrients. Conversely, France, in 2014, evidencing the highest percentage contribution of calcium and folate, did not report the highest consumption of the same minerals.

Calculations of the total fiber and nutrients from all 21 studies and the percentage contributions to this from the total amounts of the main fruit, juice, legume, and vegetable groups (Appendix A) indicated that these products contribute the highest amounts of nutrients and fiber. Tomatoes, legumes, citrus fruits and orange juice, apples and apple juice, carrots, and cabbages were of particular relevance as averaged for EFSA European elderly population data. These products are important contributors to the average daily diets of independently living elderly individuals and collectively contributed 41.1% of the fiber from all fruit, juice, vegetables, and legumes in these studies, and 47.8% of potassium, 39.8% of calcium, 47.4% of magnesium, 50.3% of phosphorus, 44.6% of iron, 40.5% of copper, 47.8% of zinc, 44.6% of manganese, 51.7% of selenium, 68.3% of RAE, 45.9% of vitamin E, 30.7% of vitamin K, 60.8% of thiamine, 42.4% of riboflavin, 47.7% of niacin, 43.6% of vitamin B6, 45.5% of folate, 46.0% of pantothenate, and 48.8% of vitamin C.

However, we noted that the nutrient calculations were based on available nutrient conversion data for each parameter and that there were gaps and assumptions (see Methods).

### 3.5. Caloric Intake from Fruits and Vegetables

When comparing the overall fruit and vegetable caloric intakes across European countries, we observed that Germany and Romania had the highest consumption (Figure 2). No differences were evident, according to the model estimation, in the fruit and vegetable intakes of elderly subjects compared to the normal population.

Concerning vegetable products, the greatest consumption was observed in Romania, Italy, and France. Sweden, Austria, and Germany showed the lowest intake (Figure 3). No differences were demonstrated across age classes.

The greatest fruit-related caloric intakes were observed for Estonia, Germany, and Denmark; conversely, Ireland and the United Kingdom showed the lowest fruit intakes (Figure 4). Considering the model estimation results, elderly people were significantly greater fruit consumers in Europe than adults overall.

Germany, Denmark, and Austria consume the most fruit and vegetable juices among European countries; on the other hand, the lowest consumption was observed for Hungary, Italy, Romania, and the Czech Republic (Figure 5). The caloric intake provided by fruit and vegetable juices was significantly greater in the overall adult population than in the elderly population.

The trace plots show the convergence of the Bayesian multilevel hierarchical model; no patterns were revealed in MCMC iterations across chains (Appendix A).

## 4. Discussion

In this study, we evaluated the fruit, vegetable, and legume consumption in 17 European countries by screening 21 studies from the EFSA database for elderly (65–74 years) citizens. Based on these data, we investigated the contributions of these foods towards achieving the established nutrients’ RDAs/AIs and relative caloric intake. For the main (micro)nutrients, RDAs/AIs were scarcely fulfilled by the non-animal-derived part of the elderly diet. Vitamin C was, as expected, the only micronutrient that reached an adequate intake. For the considered studies, 43% of them reported an adequate (>400 g/day) consumption of fruit and vegetables in elderlies, but we also found a peculiar pattern in consumption with respect to the average adult population. Not one of the countries reached the 600 g/day consumption threshold.

Interestingly, considering caloric intake, we found no differences in the dietary habits of elderly populations concerning overall fruit and vegetable consumption. Differences were instead evident when foods were divided into subclasses. The elderly populations were more likely to consume fruits and fruit products concerning the overall adult population but were less likely to drink fruit and vegetable juices and nectars. We speculate that this could be due to the usual reluctance of elderly people towards beverages. Moreover, since juices are perceived as a “sugar treat” and less healthy than the corresponding whole fruit, the constant rise in diabetes prevalence among elderly populations could have been a disincentive towards consumption.

Higher consumption of F&V has been associated with a lower risk of all-cause mortality [39], cardiovascular mortality [39], major CVD, myocardial infarction, non-cardiovascular mortality [40], CHD, stroke, and cancer [41]. Some meta-analyses findings have supported public health recommendations to increase F&V intake to up to 800 g/d for the prevention of CVD and premature mortality [41], although others have reported benefits maxing out at approximately 375–500 g/day for both non-cardiovascular mortality and total mortality [40]. It was recently suggested that current comparative risk assessments might significantly underestimate the protective associations of F&V intake, and existing recommendations for F&V intake, which are often taken as the WHO recommendation of at least 400 g/d [12], are supported by some [42]. However, in 2019, the Global Burden of Disease defined the optimal mean intakes (and range) of (i) fruit (excluding juice) as 250 g/d (200–300 g/d); (ii) vegetables (excluding legumes, juice, pickled vegetables, and starchy vegetables) as 360 g/d (290–430 g/d); and (iii) legumes as 60 g/d (50–70 g/d), thereby totaling 670 g/d [2].

The WHO CINDI report recommended in 2000 that countries should aim to have 600 g/person/day of F&V available, but food balance sheets indicate that this is not the case for many countries [12].

Studies have concluded that low consumption of F&V, particularly in low-income countries, is associated with unaffordability [43]. Low F&V consumption has been shown to decrease with increased income [44], and higher-income is associated with increased quantity and variety of F&V consumed [45]. Additional factors for low-income countries include the fact that technologies are not necessarily abundant to store or preserve perishable fresh produce to increase its availability [46]. The seasonal availability of food has been identified as being potentially significant for low-income groups who are unable to afford out of season F&V [47]. It has been suggested that increasing income and/or reducing prices would likely increase fruit intake globally. However, vegetable intake may not significantly increase with higher incomes, and the intake of some plant-based foods (beans/legumes, nuts/seeds) might decrease in some regions [48]. The potential increase in fruit intake with reduced prices is considered to result in distinct benefits for men and women of all ages and across most countries [48].

A US study showed that incentives giving a 30% subsidy on F&V could potentially prevent 1.93 million CVD events and 0.35 million CVD deaths and save $40 billion in healthcare costs over a lifetime. This increases to the prevention of 3.28 million CVD cases, 0.62 million CVD deaths, and 0.12 million diabetes cases, and savings of $100 billion in healthcare costs when other healthful foods are included in the incentive [49]. This emphasizes the potential benefits of increased effective policy actions on F&V intake.

Increasing the affordability of healthier foods was adopted as a key strategy (as part of policy options for promoting a healthy diet within objective 3 in the WHO 2013–2020 global action plan for the prevention and control of NCDs) [50]. The action plan specifically states (pages 32 and 33) that member states should consider developing or strengthening national food and nutrition policies and action plans, including developing guidelines, recommendations, or policy measures to increase the availability, affordability, and consumption of F&V; policy measures to engage with food retailers and caterers; promotion of the provision and availability of healthy food in all public institutions; the promotion of nutrition in educational institutions; employing economic tools, such as national taxes and subsidies; the promotion of nutrition labeling; and informing consumers through evidence-informed public campaigns about healthy dietary practice [50]. Other studies have recommended that health promotion programs and food policies should encourage healthier food choices among those in lower socio-economic positions and among those with economic difficulties in particular [45].

We reported that calculating the difference between the mean availability of fruits, vegetables, and legumes in European cluster zones and mean consumption for European elderly populations highlights the potentially residually available nutrition as percentages to meet US RDA or AIs for male adults. We tried to find the closest alternative to some of the products; however, there will indeed be some differences in nutritional values between unprocessed fruits and fruit juices. This is attributed to the lack of available data, and it is a limitation of our study.

The data show that although the lowest consumption of total fruits, vegetables, and legumes is generally associated with the lowest nutrient and fiber intake, the reverse is not always the case. The highest levels of nutrients appear to be largely associated with certain products that are consumed in relatively high quantities in some studies. This is particularly the case for Germany and Italy, where a relatively high consumption of apple juice and preserved tomatoes, respectively, contributes considerably to nearly all 19 nutrients assessed here (except folate, vitamin K, and selenium). Additionally, Romania and France, the higher vegetable product consumers, contribute at the greater part of the considered nutrients, especially fiber, RAE, and manganese.

This highlights how it is the types of products concerning different nutrition provided rather than simply the quantities of the fruits, vegetables, and legumes consumed.

This information could be helpful for recommendations for both complete and select populations, such as the elderly population, whereby higher intake of a particular fruit, vegetable, or type of legume could favor an increase in meeting total nutritional requirements while simultaneously reducing the risk of NCDs. We showed that 43% of the examined studies involving elderly populations resulted in mean consumption data below the WHO recommended daily intake (as showed in Figure 1). Interestingly, the overall intake masks the contribution of a single category of food that is different among the European countries. Knowing these data, cultural differences could be exploited to help the elderly meet nutrients recommendations through specific food valorization instead of using a generalized message.

Furthermore, since apple juice and preserved tomatoes are important and are both non-fresh products, availability concerns for low-income countries (and population groups) could be partially overcome by making such products more available and promoted, for example, at a subsidized cost.

The results of our study should be interpreted in light of some strengths and limitations. Availability and consumption data were systematically gathered and elaborated in a robust Bayesian framework to obtain strong and usable data.

### Study Limitations

A relatively small number of available studies were considered for the analysis. Moreover, similar food items without missing nutrients would be needed to increase the robustness of the cross-country F&V intake comparisons.

Another limitation consists of the exclusion, among the elderly, of the population aged more than 74 years.

## 5. Conclusions

Fruits, vegetables, and legumes are key dietary ingredients to reach a healthy old age and are important nutritional sources to maintain a healthy lifestyle in the growing elderly population. We found that elderlies in Europe have a mixed attitude toward fruit and vegetable consumption that is not fully comparable with the general adult population. This fact has to be considered when evaluating the effectiveness of penetration of food-related health policies and when comparing the results with standardized goals of RDAs or similar indices. We recommend further and regularly updated studies to be the basis of effective interventions to ease the burden of healthcare systems through healthy nutrition principles.

## Figures and Tables

**Figure 1 nutrients-12-02746-f001:**
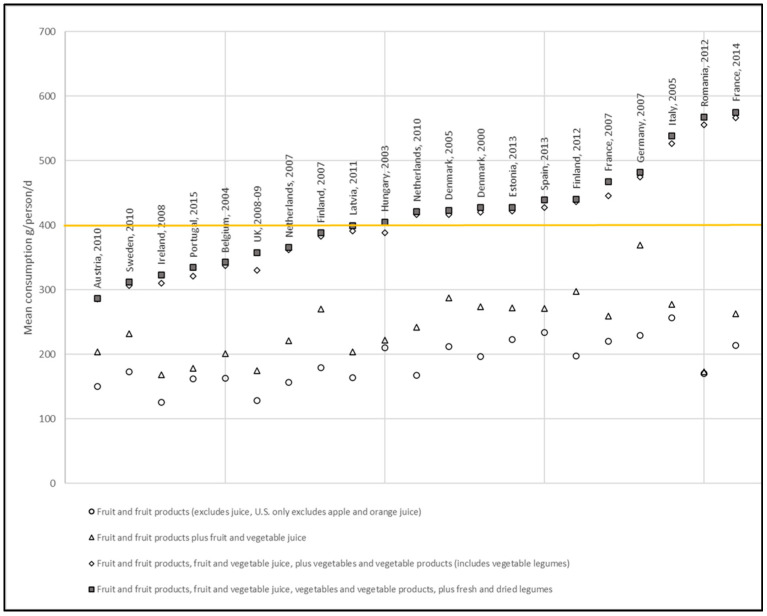
EFSA elderly (65–74 years) mean consumption data. Data for fruit and fruit products, fruit and vegetable juice, vegetables and vegetable products (including podded vegetable legumes, soya beans, and herbs), and legumes (excluding podded vegetable legumes, including dried herbs) are shown in grams/person/day. The WHO recommended daily adult intake of fruits and vegetables (excluding starchy root/tubers, but including legumes up to 80 g/d, and up to ~1 serving/150 mL 100% juice) of 400 g/d is shown by the yellow line. EFSA data are taken as chronic consumption (all subjects, g/d) at hierarchy level 1 for fruit and fruit products, vegetables and vegetable products (includes vegetable legumes), and starchy roots/tubers, and at level 2 for legumes (level 3 for legumes, fresh seeds (beans, peas, etc. without pods), or pulses (dried legume seeds)).

**Figure 2 nutrients-12-02746-f002:**
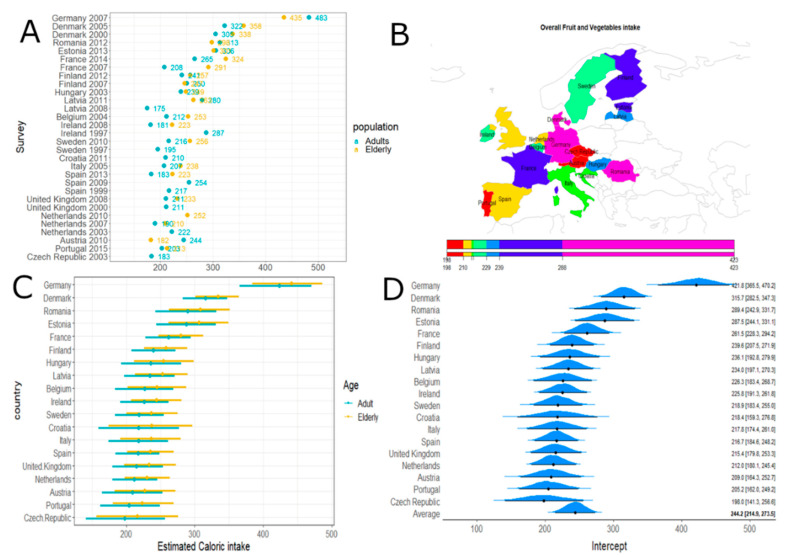
Overall fruit and vegetable caloric intake. Panel (**A**): Overall fruit and vegetable caloric intake (obtained by summing the means over the L7 levels) according to surveys and age classes. Panel (**B**): Estimated caloric intake across European countries. Panel (**C**): Estimated caloric intake across European countries according to age classes with 95% credibility intervals. Average effect estimate (95% CI) 244.18 [214.94; 273.52] and age effect (95% CI) 18.09 [–0.41; 37.43]. Panel (**D**): Estimated caloric intake across European countries (without age effect) with 95% credibility intervals.

**Figure 3 nutrients-12-02746-f003:**
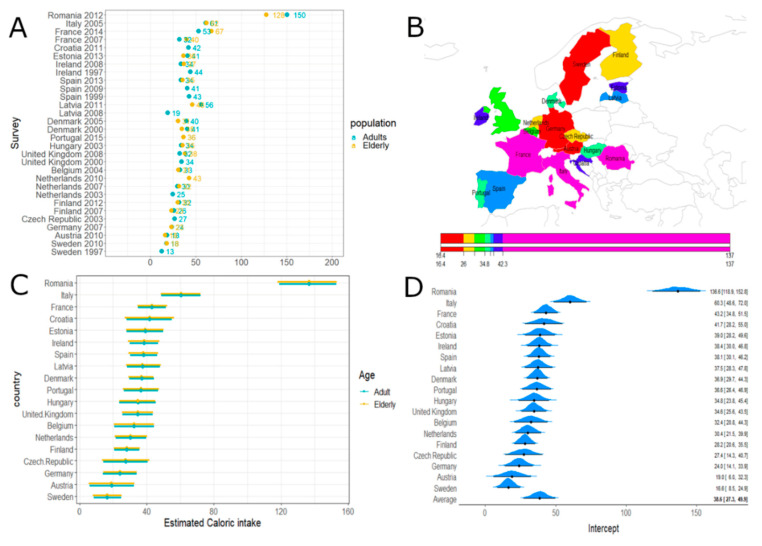
Vegetable caloric intake. Panel (**A**): Vegetable and vegetable products’ caloric intake (obtained by summing the means over the L7 levels) according to surveys and age classes. Panel (**B**): Estimated caloric intake across European countries. Panel (**C**): Estimated caloric intake across European countries according to age classes with 95% credibility intervals. Average effect estimate (95% CI) 38.59 [27.25; 49.90] and age effect (95% CI) 2.48 [–4.97; 4.83]. Panel (**D**): Estimated caloric intake across European countries (without age effect) with 95% credibility intervals.

**Figure 4 nutrients-12-02746-f004:**
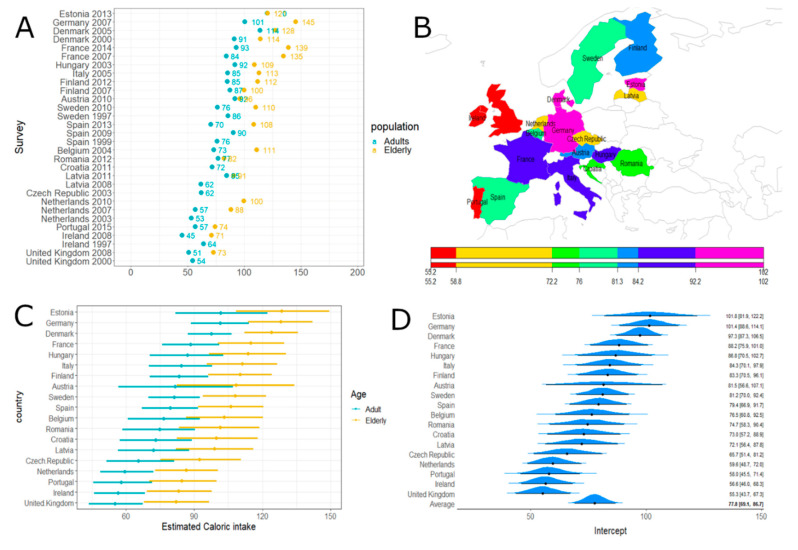
Fruit and fruit product caloric intake. Panel (**A**): Fruit and fruit product caloric intake (obtained by summing the means over the L7 levels) according to surveys and age classes. Panel (**B**): Estimated caloric intake across European countries. Panel (**C**): Estimated caloric intake across European countries according to age classes with 95% credibility intervals. Average effect estimate (95% CI) 77.8 [69.1; 86.73] and age effect (95% CI) 26.76 [17.45; 35.66]. Panel (**D**): Estimated caloric intake across European countries (without age effect) with 95% credibility intervals.

**Figure 5 nutrients-12-02746-f005:**
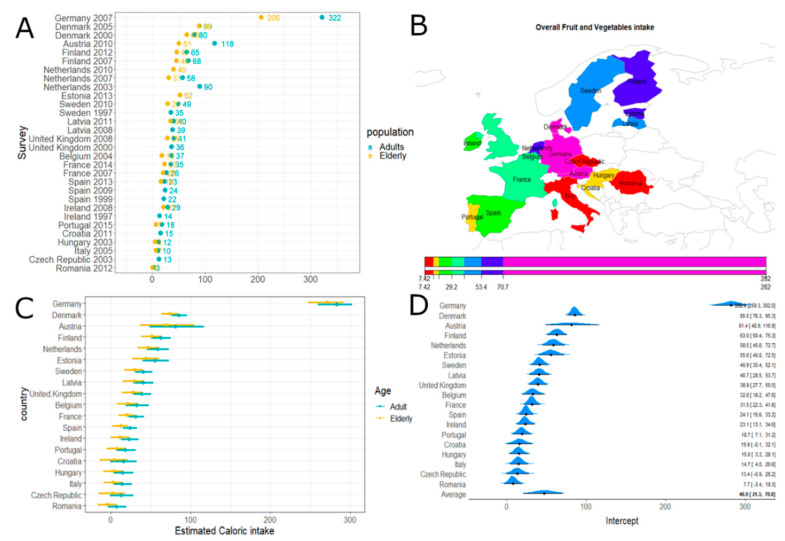
Fruit and vegetable juice and nectar caloric intake. Panel (**A**): Fruit and vegetable juice and nectar (including concentrates) caloric intake (obtained by summing the means over the L7 levels) according to surveys and age classes. Panel (**B**): Estimated caloric intake across European countries. Panel (**C**): Estimated caloric intake across European countries according to age classes with 95% credibility intervals. Average effect estimate (95% CI) 46.86 [21.27; 70.81] and age effect (95% CI) –11.62 [–5.56; –18.78]. Panel (**D**): Estimated caloric intake across European countries (without age effect) with 95% credibility intervals.

**Table 1 nutrients-12-02746-t001:** Food type availability divided by class and geographical zone.

GEMS G Code/Cluster	EFSA Countries	Fruit g/c/d	Vegetables g/c/d	Legumes g/c/d	Total g/c/d
G07	Finland, France, UK	293.3	271.3	19.7	584.3
G08	Austria, Germany, Spain	252.9	321.6	13.7	588.2
G10	Estonia, Italy, Latvia	258.1	348.8	19.3	626.2
G11	Belgium, Netherlands	336.1	288.5	28.2	652.8
G15	Denmark, Hungary, Ireland, Romania, Portugal, Sweden	246.9	354.3	18.9	620.1
Countries Mean		277.5	316.9	20.0	614.3

Availability of fruit (including fruit used for juice), vegetables (including vegetables used for juice, podded vegetable legumes, herbs, and soya beans), and fresh and dried legumes expressed as grams/capita/day (g/c/d) by GEMS geographical zone for countries with consumption data used in this study. Geographical zones correspond to countries determined to have statistical similarities in food availability (all foods) (based on FAO food balance sheets, FBS). Data taken from the IEDI database were based on the WHO GEMS/Food consumption cluster diets, with individual products initially assigned and grouped to be in general agreement with EFSA hierarchy levels 1–3 (juice is included with fruits and vegetables as unprocessed products).

**Table 2 nutrients-12-02746-t002:** US Recommended dietary allowances (RDAs) or adequate intakes (AIs) of nutrients for people >70 years old.

	Units	Male >70 Years	Female >70 Years
Dietary fiber	g/d	30 *	21 *
Potassium	mg/d	4700	4700
Calcium	mg/d	1200 *	1200 *
Magnesium	mg/d	420	320
Phosphorus	mg/d	700	700
Iron	mg/d	8	8
Copper	mg/d	0.9	0.9
Zinc	mg/d	11	8
Manganese	mg/d	2.3 *	1.8 *
Selenium	μg/d	55	55
Vitamin A (as Retinol Activity Equivalents, RAE)	μg/d	*900*	*700*
Vitamin C	mg/d	90	75
Vitamin E (as Tocopherol)	mg/d	15	15
Vitamin K	μg/d	120 *	90 *
Thiamine (Vitamin B1)	mg/d	1.2	1.1
Riboflavin (Vitamin B2)	mg/d	1.3	1.1
Niacin (Vitamin B3) (as niacin equivalents)	mg/d	16	14
Vitamin B6	mg/d	1.7	1.5
Pantothenic acid (Vitamin B5)	mg/d	5 *	5 *
Folate (B9) (as dietary folate equivalents, DFE)	μg/d	400	400

Recommended dietary allowances (RDAs) or adequate intakes (AIs, if followed by an asterisk *) of nutrients for elderly (>70 years) people. An RDA is the average daily dietary intake level sufficient to meet the nutrient requirements of nearly all (97–98%) healthy individuals in a group and is calculated from an estimated average requirement (EAR). If sufficient scientific evidence is not available to establish an EAR and thus calculate an RDA, an AI is usually developed.

**Table 3 nutrients-12-02746-t003:** IEDI cluster diet database estimating the daily availability of fiber and nutrients from total fruits, vegetables, and legumes. Estimated data were determined and summed from the mean availability of each individual product in the IEDI cluster diet database and corresponding nutrient data in food conversion databases for geographical cluster zones corresponding to countries with consumption data used in this study. Percentage contributions of estimated daily availability of fiber and nutrients from fruits, vegetables, and legumes towards the US recommended dietary allowances (RDAs) or adequate intakes (AIs *) are shown for males and females aged >70 years.

G Cluster	Availability Total F+V+L Mean (g/person/d)	Energy (kcal/d)	Dietary Fiber (g/d) *	Potassium (mg/d) *	Calcium (mg/d) *	Magnesium (mg/d)	Phosphorus (mg/d)	Iron (mg/d)	Copper (mg/d)	Zinc (mg/d)	Manganese (mg/d) *	Selenium (µg/d)	RAE (µg/d)	Vitamin E (mg/d)	Vitamin K1 (µg/d) *	Thiamine (mg/d)	Riboflavin (mg/d)	Niacin (mg/d)	Vitamin B6 (mg/d)	Folate (µg/d)	Pantothenate (mg/d) *	Vitamin C (mg/d)
**G07**	584.3	202.1	7.9	1135.8	120.2	63.5	163.1	2.4	0.3	1.1	0.8	2.8	278.7	1.6	84.1	0.6	0.2	3.0	0.4	144.8	1.5	120.4
**G08**	588.2	198.3	7.8	1180.1	129.5	62.4	152.6	2.3	0.3	1.0	0.8	2.8	280.1	1.6	116.1	0.5	0.2	2.5	0.5	151.0	1.4	118.6
**G10**	626.2	277.3	11.4	1295.7	229.8	90.7	249.0	4.2	0.3	1.7	1.0	3.3	237.2	1.8	118.7	0.6	0.2	2.9	0.5	175.9	1.4	127.1
**G11**	652.8	212.2	8.5	1285.8	137.1	66.5	169.3	2.5	0.3	1.2	1.0	4.0	426.2	1.6	84.5	0.5	0.2	2.7	0.5	164.8	1.7	118.6
**G15**	620.1	222.3	9.9	1278.7	138.6	73.4	179.7	2.6	0.3	1.1	0.8	2.7	283.3	2.0	125.2	0.6	0.2	2.9	0.5	166.6	1.4	166.0
***U.S. RDAs (AI *) males >70 y***			*30 **	*4700 **	*1200 **	*420*	*700*	*8*	*0.9*	*11*	*2.3 **	*55*	*900*	*15*	*120 **	*1.2*	*1.3*	*16*	*1.7*	*400*	*5 **	*90*
**G07**	584.3	202.1	26%	24%	10%	15%	23%	30%	29%	10%	34%	5%	31%	11%	70%	50%	14%	19%	26%	36%	29%	134%
**G08**	588.2	198.3	26%	25%	11%	15%	22%	28%	34%	9%	34%	5%	31%	10%	97%	41%	13%	15%	29%	38%	28%	132%
**G10**	626.2	277.3	38%	28%	19%	22%	36%	53%	38%	15%	42%	6%	26%	12%	99%	51%	15%	18%	27%	44%	28%	141%
**G11**	652.8	212.2	28%	27%	11%	16%	24%	31%	34%	11%	44%	7%	47%	11%	70%	43%	16%	17%	28%	41%	34%	132%
**G15**	620.1	222.3	33%	27%	12%	17%	26%	32%	33%	10%	36%	5%	31%	13%	104%	52%	14%	18%	32%	42%	29%	184%
***U.S. RDAs (AI *) females >70 y***			*21 **	*4700 **	*1200 **	*320*	*700*	*8*	*0.9*	*8*	*1.8 **	*55*	*700*	*15*	*90 **	*1.1*	*1.1*	*14*	*1.5*	*400*	*5 **	*75*
**G07**	584.3	202.1	38%	24%	10%	20%	23%	30%	29%	13%	43%	5%	40%	11%	93%	55%	16%	21%	30%	36%	29%	161%
**G08**	588.2	198.3	37%	25%	11%	20%	22%	28%	34%	12%	44%	5%	40%	10%	129%	45%	16%	18%	32%	38%	28%	158%
**G10**	626.2	277.3	54%	28%	19%	28%	36%	53%	38%	21%	54%	6%	34%	12%	132%	56%	18%	21%	31%	44%	28%	169%
**G11**	652.8	212.2	41%	27%	11%	21%	24%	31%	34%	15%	56%	7%	61%	11%	94%	47%	19%	19%	32%	41%	34%	158%
**G15**	620.1	222.3	47%	27%	12%	23%	26%	32%	33%	14%	46%	5%	40%	13%	139%	56%	17%	20%	36%	42%	29%	221%

**Table 4 nutrients-12-02746-t004:** EFSA database estimating daily availability of fiber and nutrients from total fruits, vegetables, and legumes. Estimated daily elderly intake of fiber and nutrients from total fruits, vegetables, fruit and vegetable juices, and legumes (determined and summed from the mean consumption of each individual product at level 7 in the EFSA database (65–74 years) and corresponding nutrient data in food conversion databases). The US recommended dietary allowances (RDAs) or adequate intakes (AIs *) are shown for males and females aged >70 years.

Country EFSA Database	Year	G Cluster	Consumption Total F+V+J+L mean (g/p/d)	Energy (Kcal/d)	Dietary Fiber (g/d) *	Potassium (mg/d, from g/d) *	Calcium (mg/d) *	Magnesium (mg/d)	Phosphorus (mg/d)	Iron (mg/d)	Copper (mg/d, from ug/d)	Zinc (mg/d)	Manganese (mg/d) *	Selenium (µg/d)	RAE (µg/d)	Vitamin E (mg/d)	Vitamin K1 (µg/d) *	Thiamine (mg/d)	Riboflavin (mg/d)	Niacin (mg/d)	Vitamin B6 (mg/d)	Folate (µg/d)	Pantothenate (mg/d) *	Vitamin C (mg/d)
**U.S. RDAs or AI *: males >70 y**					*30 **	*4700 **	*1200 **	*420*	*700*	*8*	*0.9*	*11*	*2.3 **	*55*	*900*	*15*	*120*	*1.2*	*1.3*	*16*	*1.7*	*400*	*5 **	*90*
**U.S. RDAs (AI *) females >70 y**					*21 **	*4700 **	*1200 **	*320*	*700*	*8*	*0.9*	*8*	*1.8 **	*55*	*700*	*15*	*90 **	*1.1*	*1.1*	*14*	*1.5*	*400*	*5 **	*75*
**Austria**	2010	G08	287.2	164.3	3.0	543.7	51.1	30.4	55.6	1.0	0.1	0.3	0.5	0.9	59.5	0.7	51.9	0.2	0.1	1.0	0.2	52.3	0.5	68.9
**Belgium**	2004	G11	342.2	165.2	4.5	668.2	64.3	34.4	80.1	1.1	0.2	0.4	0.4	1.6	117.5	1.1	48.7	0.3	0.1	1.4	0.2	75.0	0.6	59.4
**Denmark**	2000	G15	427.3	220.5	5.1	848.1	76.5	44.8	90.3	1.5	0.2	0.5	0.5	0.9	182.1	1.3	37.8	0.5	0.1	1.8	0.3	74.5	0.8	78.0
**Denmark**	2005	G15	423.0	253.3	4.7	885.2	73.1	46.4	94.8	1.5	0.2	0.5	0.6	1.3	167.9	1.1	38.6	0.4	0.1	1.7	0.3	74.2	0.8	79.6
**Estonia**	2013	G10	427.4	219.6	5.4	794.6	82.0	42.0	87.1	1.7	0.2	0.5	0.5	1.1	137.0	0.9	30.3	0.3	0.1	1.5	0.3	68.1	0.8	70.9
**Finland**	2007	G07	388.6	182.7	4.5	637.1	69.1	37.8	81.4	1.3	0.1	0.5	0.4	0.8	101.6	0.9	29.9	0.4	0.1	1.5	0.2	68.7	0.8	93.4
**Finland**	2012	G07	440.7	199.2	4.8	720.5	72.3	41.1	85.9	1.2	0.2	0.5	0.5	0.8	165.2	1.1	37.1	0.5	0.1	1.6	0.3	86.6	0.9	97.2
**France**	2007	G07	467.2	241.7	8.0	954.9	124.8	57.7	148.4	3.7	0.2	1.0	0.8	4.3	188.7	1.3	49.1	0.4	0.1	1.9	0.3	118.2	1.0	70.9
**France**	2014	G07	575.3	240.7	7.8	1145.4	144.6	61.0	145.6	2.4	0.3	0.9	0.7	2.7	203.5	1.9	75.9	0.5	0.2	2.3	0.4	148.4	1.1	88.0
**Germany**	2007	G08	481.5	390.0	4.8	1202.7	97.3	64.0	123.5	2.8	0.2	0.6	0.9	1.9	102.0	1.1	43.5	0.3	0.1	2.0	0.3	69.0	0.9	109.3
**Hungary**	2003	G15	404.7	171.8	6.4	663.6	86.6	40.8	104.1	1.8	0.2	0.6	0.4	1.3	97.6	1.0	60.1	0.4	0.1	1.4	0.3	71.6	0.7	64.5
**Ireland**	2008	G15	323.3	142.6	4.3	669.2	66.6	37.6	93.5	1.3	0.2	0.6	0.4	1.8	186.8	1.1	48.1	0.4	0.1	1.6	0.3	80.3	0.8	61.5
**Italy**	2005	G10	538.6	197.3	7.0	1416.1	130.2	81.9	159.5	2.7	0.4	1.0	0.6	3.1	308.6	3.4	101.1	0.7	0.2	3.5	0.4	144.0	1.3	109.8
**Latvia**	2011	G10	399.4	178.2	6.2	751.0	112.0	44.3	92.1	2.4	0.2	0.6	0.7	1.0	188.8	1.1	45.6	0.3	0.1	1.5	0.3	100.6	0.8	80.5
**Netherlands**	2007	G11	365.4	154.5	4.4	676.5	77.2	38.7	83.0	1.3	0.2	0.5	0.5	1.7	124.3	1.1	87.3	0.4	0.1	1.5	0.3	98.9	0.8	74.8
**Netherlands**	2010	G11	420.4	186.3	5.4	796.3	91.3	44.8	100.2	1.7	0.2	0.5	0.5	1.9	153.1	1.4	89.8	0.4	0.1	1.6	0.3	112.8	0.9	82.8
**Portugal**	2015	G15	334.3	142.3	4.3	701.2	82.8	44.9	99.3	1.7	0.2	0.6	0.5	1.3	151.0	1.5	28.1	0.4	0.1	1.4	0.3	80.3	0.7	60.6
**Romania**	2012	G15	567.5	221.8	7.8	1152.7	115.1	57.8	140.1	2.1	0.3	0.8	0.6	2.1	273.3	2.1	49.8	0.4	0.1	2.4	0.4	113.2	1.0	80.2
**Spain**	2013	G08	439.2	186.5	6.0	815.9	83.8	50.3	111.3	1.7	0.3	0.7	0.6	2.9	110.9	1.5	57.9	0.5	0.1	2.0	0.3	106.3	0.9	93.2
**Sweden**	2010	G15	312.2	163.7	3.5	559.4	49.0	32.5	67.2	1.0	0.1	0.4	0.4	0.8	89.3	0.9	21.4	0.3	0.1	1.3	0.2	60.0	0.6	66.8
**United Kingdom**	2008	G07	357.5	171.8	5.8	783.0	76.9	47.7	120.8	1.8	0.2	0.8	0.7	6.6	160.2	1.5	57.7	0.4	0.1	1.9	0.4	93.1	0.9	64.8
**Mean of EFSA studies**			415.4	199.7	5.4	827.9	87.0	46.7	103.0	1.8	0.2	0.6	0.6	1.9	155.7	1.3	51.9	0.4	0.1	1.7	0.3	90.3	0.8	78.8

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
