# Peer review of "Nutrients and Caloric Intake Associated with Fruits, Vegetables, and Legumes in the Elderly European Population"

_nutrients, 2020, doi:10.3390/nu12092746_

Round 1

Reviewer 1 Report

Abstract: I think you can strengthen the abstract by removing the last two sentences or modifying them so they are specific to your study. As they currently read, it's tough to know how your study relates.

Introduction: This is an adequate summary of the state of the literature and how your study fits within the scope of what's known.

Methods: Why did you select 4 chains and 2000 iterations?

Results:

  1. Line 237 has an error code - please fix.
  2. Figure 1 is difficult to read. It did not reproduce well. Also, the filled square boxes and open diamonds are not defined.
  3. Table 3 - the first column label is partially missing.
  4. You hop between saying the data is "consumption" and "availability." Please accurately reflect the data being presented.
  5. You also present data as "per capita" and "per person". Is there a reason why you use both?
  6. line numbers start over again on pg 11 and the font size dramatically increases in 3.5.
  7.  

Table S5 - why are there yellow highlights in the table?

Discussion: This effectively puts the findings in context. Citations are appropriate.

Author Response

Reviews attached

Reviewer 2 Report

While the paper is well written and extensive analysis has been undertaken, the purpose of the paper and conclusion is not entirely clear. The nutrients and caloric intake associated with fruits, vegetables and legumes in older Europeans has been estimated but it doesn't answer the question for the reader - do older Europeans consume enough fruits, vegetables and legumes to meet nutrient recommendations? and If not, how can these recommendations be met?

Abstract

How does the comparison of the nutrient intake of the older and general population contribute? Focus on the results for the elder population.

Introduction

This is clear and describes the benefits of fruit, vegetable and legume consumption. The paragraph (lines 78-89) discussing elder populations should focus on European/Wester/high income country populations since this is the focus of the paper. 

In the introduction, define the age group (65-75). As many people live past 75 years, be clear that the very old are not included.

Methods

The methods are comprehensively detailed. 

Why are the US RDAs and AIs used rather than EFSA Dietary reference values?

2.3: It would be useful to describe the type of dietary assessment methods used for the food consumption surveys - did most surveys use 24 hour recalls or dietary records?

202: For foods with missing nutrients, where nutrients imputed from similar foods, or is there missing data and therefore an underestimation of the nutrient intake?

210: Is it reasonable to use the unprocessed equivalent of a fruit for juice? I assume some nutrients are partially lost in the processing.

Results:

237: Add reference

Figure 1: Please check that this is the correct version. The y axis title is not readable. What are the grey boxes indicating?

Table 4: Is this availability or consumption data?

3.4 First paragraph: Be clear that the kcal/person/d are from fruit, vegetables and legumes.

Second paragraph: I found this difficult to read. I think you need to be pedantic to be clear to the reader. low percentage contributions of x had the lowest consumption of x etc.

16-18: Is this sentence incomplete? or does 'that' need to be removed from line 18 (indicated the products ...)

3.5: font size

Figure 4: Can legumes be separated from nut and oilseed. I would think much of the UK contribution is from oilseeds as to my knowledge legumes and nuts consumption is low. I don't think this figure is very helpful since we are only interested in legumes in this paper. And there are already a lot of figures.

Discussion

In the first paragraph state the key finding. What is the intake of fruit, veg and legumes? What are the implications of this?

115- which countries in this study have reached the goal of having 600g F&V available?

Para 118-129: Keep this paragraph focused on European/Western/high income countries as these are the countries in your results

Include a paragraph on the strengths and limitations. The strengths are the use of availability and consumption data but there are limitations with both of these that need to be described.

Are there more items that can be highlighted other than 2 items from 2 countries?

Conclusion

How does this conclusion relate to the study - why constantly updated? what are the policies that are being monitored? Keep conclusion related to the findings of this study - nutrients related to consumption and availability, what this means for older people in Europe and what can be done.

Conflicts of interest: Please state that two of the researchers are involved in companies that sell dietary supplements.

Table S5: Are the yellow highlights and red font indicating something?

Author Response

Reviews attached
